# Volatile Profile of Strawberry Fruits and Influence of Different Drying Methods on Their Aroma and Flavor: A Review

**DOI:** 10.3390/molecules28155810

**Published:** 2023-08-01

**Authors:** Doaa Abouelenein, Laura Acquaticci, Laura Alessandroni, Germana Borsetta, Giovanni Caprioli, Cinzia Mannozzi, Riccardo Marconi, Diletta Piatti, Agnese Santanatoglia, Gianni Sagratini, Sauro Vittori, Ahmed M. Mustafa

**Affiliations:** 1CHemistry Interdisciplinary Project (CHIP), School of Pharmacy, University of Camerino, Via Madonna delle Carceri, 62032 Camerino, Italy; doaa.abouelenein@unicam.it (D.A.); laura.acquaticci@unicam.it (L.A.); laura.alessandroni@unicam.it (L.A.); germana.borsetta@unicam.it (G.B.); giovanni.caprioli@unicam.it (G.C.); cinzia.mannozzi@unicam.it (C.M.); riccardo.marconi@studenti.unicam.it (R.M.); diletta.piatti@unicam.it (D.P.); agnese.santanatoglia@unicam.it (A.S.); gianni.sagratini@unicam.it (G.S.); ahmed.mustafa@unicam.it (A.M.M.); 2Department of Pharmacognosy, Faculty of Pharmacy, Zagazig University, Zagazig 44519, Egypt

**Keywords:** strawberry, volatile profile, drying methods, gas chromatography, active aroma components, electronic nose sensing

## Abstract

Strawberries are the most popular berry fruit in the world, due to their distinctive aroma, flavor, and known health properties. Because volatile substances play a large role in strawberry flavor, even little alterations can have a big impact on how the fruit tastes. Strawberries are thought to have a complex aroma. Fresh strawberry fruits contain more than 360 volatile compounds, including esters, furans, terpenes, alcohols, aldehydes, ketones, and sulfur compounds. Despite having far lower concentrations than esters, terpenoids, furanones, and sulfur compounds, all have a considerable impact on how people perceive the aroma of strawberries. With a focus on the active aroma components and the many analytical methods used to identify them, including gas chromatography, electronic nose sensing, and proton-transfer- reaction mass spectrometry, the present review’s aim was to provide a summary of the relevant literature. Additionally, strawberry fruits are frequently dried to create a powder in order to increase their shelf life. Consequently, the impact of various drying techniques on strawberries’ volatile profile was investigated in the current review. This review can be considered a good reference for research concerning the aroma profile of strawberries. It helps to better understand the complex aroma and flavor of strawberries and provides a guide for the effects of drying processing.

## 1. Introduction

Strawberry fruit, Rosaceae family, is a delightful fruit worldwide thanks to its unique aroma and nutritional quality. Strawberries (Figure 1) gain a lot of attention due to their noteworthy biological activities and promising health benefits [1]. These properties are ascribed to phenolic compounds, vitamins, and ascorbic acid, as well as to anthocyanins, which exhibit anti-cancer and anti-inflammatory properties [2]. In addition, anthocyanins contribute to the bright red color of the fruit and to its antioxidant features. Recently, besides antioxidant and anti-inflammatory activities, polyphenols have been studied to understand their potential protective effects on cell in vitro; however, further in vivo studies are necessary to understand the observed in vitro outcomes, particularly the factors and mechanisms involved [1].

Strawberry quality and, in particular, nutritional and flavor compounds are widely influenced by preharvest factors; among them the environmental condition, soil types, fertilization, composts, and water supply, which affect the nutritional composition, the physical properties, the fruit yield, and, indeed, the fruit shelf life, as thoroughly described by Asao and Asaduzzaman 2019 [3].

The aroma of strawberries is essential for its overall appeal and plays a crucial role in its commercial success; in fact, customers purchase strawberries mainly for its special flavor [4]. The unique aroma and flavor derive from numerous volatile organic compounds (VOCs) that include esters, alcohols, ketones, furans, terpenes, aldehydes, and sulfurous compounds [5]. Despite the complex volatile profile, the fruity and floral strawberry aroma derives particularly from methyl and ethyl esters. Different strawberry varieties possess distinctive VOCs; consequently, it is possible to recognize the variety of strawberry from its volatile profile [6]. The quantitative and qualitative analyses of strawberry aroma compounds are crucial in predicting and better understanding strawberry’s key odorants [7]. Recently, various chromatographic methods, such as gas chromatography–mass spectrometry (GC–MS), proton- transfer- reaction mass spectrometry, and electronic nose, have been widely used to identify and quantify the aromatic components of strawberry fruits [8]. In the current studies, headspace solid-phase microextraction combined with gas chromatography–mass spectrometry (HS-SPME–GC–MS) was used to determine and analyze the volatile profiles of strawberries. Additionally, the relationship between volatile compounds and the E-nose sensors has been frequently explored through various statistical analyses. Finally, HS-SPME–GC–MS was also coupled with PTR-MS, for example, to study the changes during storage in the volatile profile of strawberries harvested at different maturation times [9].

To increase the acceptability and the safety, as well as improve the shelf life and the fruit quality, different drying methods, such as vacuum freeze drying (VFD), sun and shade drying, microwave, hot air drying, and oven drying, have been investigated [10,11,12]. Furthermore, to achieve these goals, great attention was given to different treatments and processes like ultrasound (US), ultrahigh pressure (UHP), and osmotic dehydration (OD) as pretreatments for drying, aimed at accelerating the drying time and guaranteeing less energy consumption and tasty flavor of the fruit snacks [11,13]. In general, the most suitable technology is selected based on the nature of the raw material and the desired features of the final products. Noteworthy, the application of a heating source could generate undesirable changes such as loss of freshness, color, texture, and aroma [14]. For this reason, it is important to select the proper drying treatment in order to ensure appreciated food products with a diversified and unique taste. In the current review, the complex aroma and flavor of strawberries have been discussed, taking into consideration the effects of the drying process on the volatile profile of *Fragaria* spp.

Strawberries and derived products have significant market importance due to their nutritional value, versatility, and wide range of applications in recipes and in the food and beverage industries. Strawberry is considered a fruit with a steadily increasing consumer demand and year-round availability of production, which is guaranteed by multiple production locations around the globe [15]. It is a fruit produced commercially in over 77 countries in the world [16]. According to FAOSTAT, in 2021, the world import quantity accounted for 1,039,444.43 tons [17]. China is the biggest producer of strawberries worldwide, with more than 4 million tons, followed by the USA, Mexico, Egypt, and Turkey [17]. The growing demand for strawberries is due to an increase in praise among consumers for their unique taste, freshness, and health benefits. Today’s consumers are more informed on the quality of food products and their health properties; thus, they are in demand for their high quality, being nutritionally versatile, and safer plant produce, which are key drivers of the market’s growth [18].

This trend is expected to further develop the global market scenario for strawberries in the coming years, with a consequent increase in the demand and production of fresh strawberries and processed strawberry products, such as jellies, jam, and syrups. Predictions on the increase in strawberries’ demand showcase that not only fresh fruits will increase the market growth but also e-commerce and online grocery shopping [19]. A growing demand for strawberries will result in an increase in crop production. In fact, strawberry productions are susceptible to abiotic and biotic factors, singly or concomitantly, such as crop pathogens, which cause significant qualitative and quantitative damages, leading to important economic losses. Therefore, sustainable agricultural strategies need to be addressed to increase the market resilience in line with this increasing market demand of areas under cultivation but with lower production and productivity rates due to disease constraints [20].

## 2. An Overview of Strawberry Aroma

Fresh strawberries, as the most consumed berry fruit in the world, are well known for their exceptional flavor derived from the combination of numerous VOCs [21,22,23]. The characteristic combination of VOCs of each strawberry variety depends not only on the cultivar genotype but also on the ripeness stage and on other pedo-climatic environment factors [24]. Systematic research of the scientific literature between 2018 and 2023 resulted in a total of nine articles reporting VOCs in red fresh strawberries (Fragraria × Ananassa). The reviewed data reported using either area percentages or internal standard response factors, leading to an impossible comparison at the quantitative level, as the results obtained in each research article were related to the specific experimental methods. A total of 181 compounds belonging to different classes, namely esters (78 compounds), terpenes (21 compounds), and acids (20 compounds), followed by alcohols (19 compounds), hydrocarbons (13 compounds), aldehydes (9 compounds), ketones (5 compounds), furanones (5 compounds), lactones (7 compounds), and aromatic compounds (4 compounds), were found in 20 specified strawberry cultivars (Albion, Calinda, Camarosa, Candonga, Candonga Sebrosa, Crystal, E-22, Elide, Festival, FL 127, Florida Fortuna, Fortuna, Frontera, Monterey, Plared, Portola, Rubygem, Sabrina, Sahara, and Victory). The complete dataset is reported in Appendix A. Among them, 31 compounds were detected in at least 10 cultivars, including 10 esters, 7 acids, 5 terpenes, 4 aldehydes, 2 furanones, 2 lactones, and 1 ketone. A summary of the most occurrent VOCs in fresh strawberries is reported in Table 1.

Esters represent the most abundant VOCs in strawberries: 78 different molecules, accounting for 43% of all VOCs, were found in the 9 articles, as above. They are reported as the major source of fruity and floral odor characteristics of *F*. × *ananassa*, and their presence could be influenced by the different strawberry fruits ripening [21]. Cozzolino et al. [22] studied the volatile profile of a Candonga cultivar at different maturation stages, and the esters were reported as mainly related to the harvest time. Moreover, Leonardou et al. [21], investigating six different strawberries genotypes, underlined that the harvest time effects on the ester concentrations overshadowed that of the genotype. These results were in accordance with Parra-Palma et al. [20], who also reported esters as the most abundant odorants in Camarosa, Monterey, Crystal, and Portola cultivars, in which the total esters constituted 75.1%, 79.7%, 65.5%, and 78.4% of the total volatiles, respectively. Butanoic acid esters were the most abundant, followed by hexanoates and acetates [23]. In line with this, the ester results were the most relevant class also in the Florida Fortuna variety investigated by Kafkas et al. [25]. revealing percentages up to 37.22%, with acetic acid methyl ester as the principal compound. Esters were the most represented class also in Sabrina and Elide cultivars at 50.4% and 71.2%, respectively [23]. In particular, methyl and ethyl esters have been reported as the most influencing contributors to strawberries’ flavors, impacting their fruity and floral aromas. Padilla-Jiménez et al. [6] investigated the volatile profile of Frontera, Festival, and Albion cultivars at different maturation stages. This chemical class was defined to be highly predominant in the characteristic aromas of all studied cultivars [26,27]. In the Albion and Festival varieties, butanoic acid methyl esters increased in concentration during the maturation of the fruits, resulting in a key compound that could be used as a maturity indicator in these varieties. Red Frontera strawberries reported the highest concentration of hexanoic acid methyl ester and isopropyl butyrate. These ester compounds are closely related to mature strawberries’ flavor and were also found in high concentrations in other mature fruits, such as peaches or grapes [28,29]. Urün et al. [30] investigated the volatile profile of ten strawberries cultivars, namely Rubygem, Fortuna, Festival, Calinda, FL127, Plated, Sahara, Sabrina, Victory, and E-22, concluding that mainly methyl and ethyl esters with a few other volatile substances strongly contributed to the strawberries’ aromas. The Camarosa, Candonga, and Festival cultivars were compared by González-Domínguez et al. [31], revealing Camarosa to be the richest cultivar in esters. Methyl butyrate and hexyl hexanoate were the most detected VOCs. Abouelenein et al. [12] focused on the drying method and strawberries-based food products, comparing their volatile profiles with fresh strawberries. Hexyl acetate and (E)-2-hexenyl acetate were the most abundant esters in fresh commercial strawberries, but their content significantly decreased after drying processes and jam preparation. The chemical class of terpenes in strawberries has been of particular interest, since these compounds confer specific sensory properties and have also been studied as molecules with biological activity against microorganisms (fungi, insects, and bacteria) [32]. Together with esters, terpenes are the most representative chemical class of strawberries’ aroma profiles.

Terpenes, despite their low concentration, may have a significant impact on preferable fruit aroma characteristics, as assessed by Kafkas et al. [28], reporting total terpenes percentages between 2.39% and 5.35%. Similarly, Urün et al. [33] found terpenes from 0.34% to 2.61%. Linalool, nerolidol, α-terpineol, α-farnesene, and geraniol were the most recurring compounds in red fresh strawberry fruits, being responsible for their floral, fruity, citrus, and green odors. In particular, linalool, and nerolidol were detected as the main volatile terpenoids in cultivated ripe strawberries, whereas α-pinene, β-myrcene, α-terpineol, β-phellandrene, and myrtenyl acetate were mostly identified in wild strawberries [5]. Linalool was the most occurrent VOC in selected research articles, and it was the most abundant in the Elide strawberry variety [26]. The Candonga and Festival varieties were richer in terpenes than the Camarosa cultivar [31]. In the Candonga cultivar, linalool, together with α-muurolene and butanoic acid methyl esters, was reported to be an indicator of fruit ripeness, as it modifies during ripening processes. The authors assessed that its concentration could offer information about the optimal harvest time, even if it seems to not be transferable to other cultivars [22,26]. Similar conclusions were drawn by Padilla-Jiménez et al. [6], who reported terpenes as the earliest maturation stage markers for the Frontera variety; in fact, they participate in the protection of the plant during its initial development. Moreover, the concentration of these molecules could be influenced by the strawberry genotype. Leonardou et al. [21] found the highest level of terpenes in the Fortuna cultivar among a total of six studied samples. Further confirmation of the impact of genotype on the content of these secondary metabolites was provided by Parra-Palma et al. [23], with Monterey the richest in terpenes when compared to the Camarosa, Crystal, and Portola varieties. Important variations in the terpenes content were observed after the drying processes and preparation of strawberry-derived products such as jams [12].

Several odor flavors can be generated by volatile organic acids, even if they represent secondary components of strawberries’ aroma [33]. The 2-methyl butanoic and hexanoic acids were the two main occurrent acids in the reviewed articles. Hexanoic acid was reported to be an off flavor responsible for some unpleasant flavors of *F*. × *ananassa* [12]. A small group with 4–5% of the reviewed compounds can be classified as aldehydes. They increase in concentration during maturation, and their occurrence is influenced by the genotype [21,26]. Increased levels of aldehydes and ketones have been observed in heat-dried strawberries and derived products, such as jam; however, they decrease after freezing [12]. Furaneol (DHMF) and mesifurane (DMMF), responsible for caramel-like, sweet, and floral aromas, are the major furans found in strawberries [24]. The contents of these furanic compounds are related to the genotype and *F*. × *ananassa* ripening time [21]. DHMF and DMMF increase during ripening in Candonga strawberries [22]. Red Festival strawberries showed a higher concentration of furaneol when compared to red Camarosa and Candonga [31]. Additionally, mesifurane is highly concentrated in Camarosa, Portola, and Monterey, while Cristal strawberries report γ-decalactone as the most abundant compound [23]. Various lactone compounds are widely described as key odorants, defining the pleasant peach-like and fruity aroma. In particular, γ-decalactone increases during harvest time [21,22].

Further chemical classes of VOCs in fresh strawberries are alcohols, hydrocarbons, ketones, and aromatic compounds, found in small percentages in all the reviewed works. In conclusion, we can assess that the aroma of *Fragraria × ananassa* fresh fruit does not come from just one or a few flavoring molecules, but it shows quite complex chemical patterns that include more than 360 volatiles in specific concentrations to create the unique and intense flavor of strawberries [32]

## 3. Effect of Different Drying Methods and Conditions on Strawberry VOCs and Aroma

The food sector places a lot of importance on drying procedures. Fresh strawberry fruits quickly lose their quality and become perishable, causing significant economic loss. Their short harvesting seasons further restrict their commercialization and consumption, because the fruits are not always available. One proposed conservation strategy to increase their shelf life and maybe boost utilization of the fruit is to dry the strawberries to create a powder in order to maximize the nutritional benefits of the fruit [34]. For the manufacturing of strawberry powders, a variety of drying techniques have been used, including oven, natural air, microwave, and freeze drying. Oven drying is a popular and affordable approach, but it exposes fruits to high temperatures and oxygen, which could alter their chemical composition. Furthermore, freeze drying is one of the greatest drying techniques for creating superior fruit powders, but it is rather expensive. Although freeze drying maintains the sensory qualities, some researchers have claimed that this method may result in the loss or alteration of bioactive substances [35,36]. One of the main goals of the current review was to compare how various drying techniques affected the volatile profiles of strawberry fruits to determine which drying technique would be most effective for producing strawberry powders with a desirable aroma, one of the most crucial quality attributes. The reported data are summarized in Table 2 and Figure 2 and further discussed below.

Abouelenein et al. [12] investigated the effects of various drying processes (oven drying at 45 °C and 60 °C, freeze drying, microwave drying, and shade air drying) for the production of strawberry powders with suitable aromas, using HS-SPME–GC–MS and principal component analysis (PCA), and 85.03–96.88% of all headspace compositions were represented by a total of 165 volatile substances. The research demonstrated that various drying techniques had varying effects on the kind and concentration of volatile chemicals in strawberries and that clear separations between the various strawberry treatments were produced. The PCA showed that hexyl acetate, ethyl hexanoate, mesifurane, (E)-2-hexenyl acetate, (E)-nerolidol, 1-hexanol, γ-decalactone, and acetoin were the most notable typical differential volatiles. According to the literature, shade air-dried, freeze-dried, and 45 °C oven-dried samples all kept more of the fruity and sweet fragrances of strawberry fruits, accounting for more than 68% of the total strength of the aroma. Contrarily, a severe loss of fruity ester volatiles occurred by microwave drying, while intermediate loss resulted in the oven-dried 60 °C sample. According to its principal aroma features, a prior study divided strawberry aromas’ active volatiles into six aroma groups, namely fruity (esters and lactones), representing more than 50% of the total aroma strength, followed by sweet (furanones, which account for roughly 18% ), floral (terpenes, which make up the least intense strawberry aroma class at 7%), green (aldehydes, 10%), volatile acids (9%), and other (miscellaneous) [33].

In freeze- dried (FD) strawberries, Al-Taher et al. [37] were able to pinpoint many volatile components and quantify significant odor-active ones. Mesifurane and furfural were the two most significant aroma compounds discovered in FD strawberries, making up 38.1% and 21.8% of the total volatiles, respectively. However, the most significant furanone found in fresh strawberries, furaneol, was not found in the FD samples. Other furanones, including gamma-decalactone (2(3H)-furanone, 5-hexyldihydro-), 3(2H)-furanone, 2(1-hydroxy-1-methyl-2-oxopropyl)-2,5-dimethyl, and 2,4-dihydroxy-2,5-dimethyl-3(2H)-furan-3-one, were found and accounted for around 15.0% of the total volatiles in FD strawberries. Esters, pyranones, lactones, and other minor substances were also found in both fresh and FD strawberries in this study. Similarly, good preservation of the aroma (fruity esters) was observed for freeze-dried strawberry paste prepared with the addition of 3% trehalose [38].

The amounts of 11 flavor important volatiles in strawberry samples were examined comparing four treatments, including control, spray drying, refractance window (RW) drying, and freeze drying, by Abonyi et al. [39]. The ratio of concentration to threshold value, known as the aroma value, was utilized by the authors to describe odor-active strawberry aroma components [40]. Nerolidol, hexanol, linalool, ethyl butanoate, ethyl acetate, methyl butanoate, and ethyl hexanoate were among the most significant flavor notes in the strawberry puree (the control) in terms of the aroma values. The fruity and green ester notes in strawberries that were RW- and spray-dried were significantly diminished when compared to the control. The freeze-dried strawberry product, in contrast, effectively preserved the fruity and green components. Although the concentration of nerolidol decreased along the sequence of freeze-dried, RW-dried, and spray-dried samples, the control had a significant nerolidol content. The relative effect of nerolidol in the RW- and spray-dried samples on the overall odor impression was considerably more noticeable when considering the reduction in fruity and green aroma components. When compared to the control and freeze-dried samples, it could be shown that the ethyl acetate content was lower in the RW- and spray-dried samples. The RW-dried samples included a high concentration of a one heat-enriched component (carvone), which has not been previously reported. In the control/spray-dried samples, the content of this substance was low, and in the freeze-dried samples, it was undetectable. It was quite difficult to understand the exact explanation behind the enrichment of the carvone content in the RW-dried samples. The authors of the same study noted an increase in the aldehyde level during RW and spray drying at high temperatures. The ketone concentration for the RW-dried samples also increased significantly. This showed that ketone- and aldehyde-rich dehydration at high temperatures (RW and spray drying) changed the overall flavor impression in the dried samples.

Tekgül et al. [41] observed that 10 volatile substances not present in fresh strawberries were found in their headspace after drying strawberries at three different temperatures. These compounds were two acids (acetic acid and isobutanoic acid), two aldehydes (2-methylbutanal and 3-methylbutanal), two Maillard reaction and sugar degradation products (2,5-dimethyl-3(2H)-furanone and 2,3-dihydro-3,5-dihydroxy-6-methyl-4H-pyran-4-one), one aromatic hydrocarbon (m-xylene), one ketone (acetoin), and one sulfur derivative (dimethyl sulfide). Hexanoic acid, limonene, and hexyl acetate in the “Osmanl” samples were degraded to levels below the limit of detection. All the components from the fresh samples of the “Florida Fortuna” samples were likewise found in the dried samples. Nonanal and m-xylene were not found in any “Osmanl” strawberry, and 2,2,4,4-tetramethyl butane and toluene were not found in any “Florida Fortuna” strawberry, according to the substances found in the dry samples. Eight common chemicals, including ethyl acetate, diacetyl, methyl butyrate, heptane, hexanal, methyl hexanoate, (E)-2-hexenal, and ethyl hexanoate, were significantly reduced in both varieties after drying. Acetone and nonanal were also greatly reduced in the “Florida Fortuna” samples, whereas toluene was significantly reduced in the “Osmanl” sample. In general, the most prominent volatiles in the dried samples were dimethyl sulfide (cabbage, sulfur, and gasoline-like odor); acetic acid (vinegar and sour odor); acetone (nail polish odor); furfural (bread, almond, and sweet odor); and acetoin (butter and cream odor). Acetone, which was the most prevalent substance in the fresh samples, was greatly reduced in the “Florida Fortuna” samples after drying, making dimethyl sulfide the most abundant compound at 50 °C. The amounts of furfural in both samples dramatically rose as the drying temperature was raised. Furfural was the most prevalent substance in the samples that underwent treatment at 60 °C and 70 °C. Dimethyl sulfide, one of the major volatiles in dried strawberries, is an off-flavor substance created during the thermal treatment of vegetables and fruits [42]. Furthermore, prolonged exposure to high temperatures may have a negative impact on strawberries’ volatile profile due to the breakdown of existing substances and the synthesis of new ones. The samples dried at 50 °C required the longest treatment time in this study. However, the findings revealed that drying at a low temperature (50 °C) led to the greatest retention of acetone—the major chemical in fresh strawberries—and the least amount of furfural production. Additionally, the samples treated at 50 °C had the maximum retention of esters, essential volatiles with fruity odors. The authors suggested that, in order to have dried strawberries with a fruitier flavor, drying at low temperatures for an extended period of time would be preferred over drying at high temperatures for a shorter period of time.

Based on the available literature, vacuum freeze drying (VFD) was investigated as a drying method to be applied after different pretreatments, such as ultrasound (US), ultrahigh pressure (UHP), or their combination [13,43]. The application of these pretreatments as new hybrid drying technologies allowed to save on the energy request, as well as obtain products with suitable quality features [44,45]. An increased number of volatile compounds have been identified by applying a pretreatment prior to the VFD in the following order: UHP-US > UHP > US [13]. In more detail, the most representative were esters (ethyl acetate, hexyl butanoate, ethyl hexanoate, methyl acetate, and methyl hexanoate; 51.22–61.43%), followed by aldehydes ((e)-2-hexenal, hexanal, heptanal, (e)-2-heptenal, and pentanal; 20.10–33.67% of the total volatiles); alcohols (hexanol; 0.32–1.45%); ketones (methyl heptanone and 1-octen-3-one; 6.62–15.55%); and others (0.37–10.80%). However, in the control samples, higher levels of esters and alcohols were found: 57.72% and 31.06, respectively. Among these, ethyl hexanoate (18.57%), methyl butyrate (16.97%), methyl hexanoate (14.77%), hexanal (10.16%), (E)-2-heptenal (8.43%), and (E)-Oct-2-enal (4.39%) were the most abundant. For UHP pretreated strawberry samples, a decrease in esters and aldehydes was observed, particularly hexyl acetate, methyl butyrate, methyl hexanoate, ethyl hexanoate, hexanal, (e)-2-heptenal, (e)-Oct-2-enal, nonanal, and (z)–2-nonenal, which are associated with the fruity, herb, sweet, wine, pineapple, apple, green, nut, and fat sensory descriptors previously identified in pears and pink guava fruit [46,47]. For the US dried strawberries samples, an increase in esters (methyl acetate, hexyl acetate, ethyl butanoate, and hexyl butanoate) was reported. Conversely, in the UHP-US dried samples, higher aldehydes compounds were found compared to the control, creating apple, green leaf, almond malt, and pungent notes in the final products. The modifications in the volatile compounds between the different VFD samples were ascribed to the US and UHP pretreatments, which provoked physical and enzymatic effects by application of the sonication and pressure, respectively [13,48,49]. Xu et al. [40], regarding the aroma of *Fragaria × ananassa* Duch, cv. HongYan determined by electronic nose analysis, observed similar response values but different signal intensities for the VFC strawberries pretreated with different US processing at 20 kHz, 40 kHz, and a simultaneous dual-frequency mode. In general, aromatic compounds, organic sulfides, and nitrogen oxides characterized the odor of VFD strawberries. A higher response signal in terms of aromatic compounds, which represent a valuable characteristic of dried fruit flavors, was highlighted for strawberries pretreated with the dual-frequency mode prior VFD. The sensor’s signal representing nitrogen oxides and their negative effects on flavor showed higher values for strawberries sonicated in the dual- frequency mode. This behavior was explained by the US effects on strawberry tissue, which promoted water removal and thus contributed to reducing the drying time [11,50].

These results offer a roadmap for the drying effects and a better understanding of strawberries’ aroma. The volatile profile of strawberries is heavily impacted by the drying process, and each drying procedure has a unique impact. Each drying method might be better suited to the creation of a certain targeted flavor food ingredient, as each strawberry powder’s volatile composition was different. The created powders could be used to flavor dairy and baked products with appealing strawberry flavors, enhancing their functional value. It is advised to conduct more research to better understand the impact of volatile and quality modifications on strawberry flavor and consumer acceptance.

**Table 2 molecules-28-05810-t002:** Effect of different drying methods and conditions on strawberry aroma and flavor.

Type of Drying	Drying Conditions	Main Findings	Ref.
Shade air drying	- Cut strawberry pieces were put on watch glasses and dried indoors in the shade for 96 h.	- Shade air drying preserved more of the sweet and fruity aromas of strawberry (furanones, lactones, and esters).	[12]
Oven drying	- Drying using an electric oven at 45 °C and 60 °C for 24 h and 16 h, respectively.	- Oven drying at 45 °C retained more of the fruity and sweet aromas of strawberry (esters, lactones, and furanones). - Oven-dried at 60 °C sample demonstrated an intermediate loss of fruity esters.	[12]
Hot air drying	- Samples were cut into. The 3 mm thick slices were dried with constant air flow (1 m/s at 50 °C for 27 h, 60 °C for 21 h, and 70 °C for 15 h). - Using a dehydrator, drying was accomplished. - When samples had a moisture level of 10 ± 1%, drying was complete.	- In dried strawberries, 24 volatiles were found. - Drying at a low temperature (50 °C) had the lowest impact on sugar degradation and the highest retention of fresh strawberry volatiles. - Florida Fortuna and Osmanl varieties of dried strawberries displayed a similar trend in their volatile profiles at high temperatures (60 °C and 70 °C). - Dimethyl sulfide, acetic acid, and acetone were the main volatiles. - Aldehydes, followed by acids, esters, and products of sugar breakdown were the majors.	[41]
Freeze drying	- Not specified	- 29 volatiles, including aldehydes, terpenes, esters, acids, furanones and alcohols, were detected in FD samples. - 2,5-dimethyl-4-methoxy-3(2H)-furanone (Mesifurane) and furfural were the two main aroma components in FD strawberry and account for 38.1% and 21.8%, respectively, of the total volatiles. - Mesifurane, was found to have the highest concentration in FD strawberries (93.25 ± 29.5 µg/g), compared to gamma-decalactone (9.48 ± 0.31 µg/g) and furfural (8.27 ± 1.06 µg/g).	[37]
	- Freeze drying at −54 °C, with a shelf temperature of 10 °C and a pressure of 0.05 mbar for 48 h.	- Freeze drying retained more of the sweet and fruity aromas of strawberry (furanones, lactones, and esters).	[12]
	- Absolute pressure was of 3.3 kPa. The temperature of the heating plate was 20 °C, while the condenser temperature was –64 °C. - The drying time was 24 h.	- The fruity and green components were well retained in the freeze-dried strawberry product. - Less nerolidol content was observed compared with the control.	[39]
Microwave drying	- Drying using microwave oven of 900 W, for 4 min.	- Microwave drying method showed drastic loss of fruity esters. - The total ester comprises 0.35% of total volatiles in microwave-dried strawberry samples, with respect to 36.96% in fresh samples.	[12]
Refractance window drying	- 0.7 m/s of air at 20 °C and 52% relative humidity was pushed over the bed. - 95 °C was the water’s temperature, and the belt’s speed ranged from 0.45 to 0.58 m/min. - Strawberry puree without the maltodextrin carrier was dried to 5.7% (wb) and strawberry puree dried to 9.9% (wb).	- The RW-dried samples showed a significant reduction in nerolidol and fruity and green ester notes in comparison to the control. - The amount of ethyl acetate in RW-dried samples was lower than in the control and FD samples. - The RW-dried samples demonstrated a high concentration of carvone. - Aldehyde content was found to be increased. - A significant increase in ketone content was noted.	[36]
Spray drying	- The inlet air temperature was 190 ± 5 °C and the outlet air temperature 95 ± 5 °C. - Strawberry puree was given a 70% maltodextrin carrier, and samples were dried to a moisture level of 2.3% (wb).	- A significant decrease in nerolidol and notes of fruit and green ester when compared to the control was produced. - The amount of ethyl acetate was lower than in the control and freeze-dried samples. - Aldehyde content was found to be increased.	[39]
Vacuum freeze drying	- Not specified	- Highest retention of fruity esters on freeze dried strawberry samples with the addition of 3% of trehalose	[38]
	- Ultrasound (US), ultrahigh pressure (UHP) and UHP-US followed by vacuum freeze drying (VFD): - cold trap T of −50 °C and p = 10 Pa, heating plate at 4 °C for a total drying time of 20 h	- 30, 35 and 47 volatile compounds have been identified respectively in US, UHP and UHP-US pretreated strawberries. The most abundant compounds were esters (51.22–61.43%) in all VFD strawberries. - Ethyl acetate, methyl butyrate, methyl hexanoate, ethyl hexanoate contributed to the pineapple, fruity, and apple- like aromas of dried strawberry slices. - 10.97% of aldehydes were lost in US and UHP pretreated strawberries.	[13]
	- Ultrasound (US) assisted vacuum freeze drying (VFD): - pre-frozen at −40 °C, drying T of 25 °C, p = 0.518 Mbar and cold trap T of −90 °C. - The final drying T of 35 °C.	- The e-nose discriminated aromatic compounds, organic sulfides, and nitrogen oxides for the VFD strawberry slices. - Dual frequency in sequential mode showed best retention in aromatic compounds.	[43]

## 4. Analytical Methods for the Determination of Aroma Compounds in Strawberry

A wide range of VOCs play an important role in assessing food spoilage, and that is why the study of VOCs in food products has been a critical issue for decades. Gas chromatography (GC), proton- transfer- reaction mass spectrometry (PTR-MS), and electronic nose (E-nose) are the most important analytical techniques used to study the volatile profile of different plants [9], as reported in Table 3.

Headspace solid- phase microextraction–gas chromatography–mass spectrometry (HS-SPME–GC–MS) is one of the most common analytical methods used to study the volatile profile of strawberries. The HS-SPME technique is one of the most common analytical approaches for VOC analysis, because it is simple, cheap, solvent-free, easy to handle, and very sensitive. In fact, this type of analysis was used in all the studies reported in the table for different purposes. Buvé et al. [51] used this analytical method to study changes in VOCs’ compositions in strawberry juices during conservation, highlighting the importance of storage conditions and temperature in the volatile profile [51]. Li et al. [52] used it to compare the volatile profiles of two cultivars of strawberry, ‘Amaou’ and ‘Yuexin’ (*Fragaria* × *Ananassa*), demonstrating that the soil culture can improve the aroma profile in cultivated strawberry ‘Amaou’ fruits [53]. In the same way, Abouelenein et al. [12] investigated the influence of freezing and different drying methods (oven drying at 45 °C and 60 °C, freeze drying, microwave drying, and shade air drying) to produce strawberry powders with desirable aromas; they also analyzed the volatile profile of commercial jams to evaluate their quality in terms of aroma components. The results showed that freezing and different drying methods exerted different influences on VOC contents in strawberries, highlighting that the shade air-dried, frozen, freeze-dried, and 45 °C oven-dried samples retained more of the fruity and sweet aromas of strawberry fruits. The strawberry jams demonstrated almost a complete destruction of the esters and alcohols in most jam samples, while the terpenes were significantly increased; in addition, off-flavor and heat-induced volatiles, such as furfural and 5-hydroxymethylfurfural, were generated [9]. On the other hand, HS-SPME–GC–MS was used by Liu et al. (2023) to find a possible influence of the volatile profile of strawberry ‘Red face’ flowers and the pollinators’ behaviors (*A. mellifera* and *B. terrestris*) [54]. González-Domínguez et al. [31] studied the volatile profiles of three different varieties of strawberries (Camarosa, Candonga, and Festival) cultivated in a soilless system with a headspace solid- phase microextraction–gas chromatography–flame ionization detector (HS-SPME–GC–FID). The results from this study showed that the genotype can influence the volatile profile of strawberries, and geraniol and hexyl hexanoate were useful in discriminating among different cultivars [31]. Cozzolino et al. [22] used headspace solid- phase microextraction–gas chromatography–triple- quadrupole mass spectrometry (HS-SPME–GC–qMS) to study the changes in fruit quality traits and the contents of VOCs and phenolic compounds in *Sabrosa* strawberry samples collected at two different ripening stages (half-red and red) at three different harvesting times and to demonstrate that the ripening stage is determinant to meet consumers’ liking, while the collection times did not influence the fruit quality [25]. Finally, Dubrow et al. used a dynamic headspace–gas chromatography–time-of-flight mass spectrometry analysis (DHS–GC–Q-ToF) to identify VOCs in strawberry preserves to find a possible correlation between the volatile profile and consumer liking; they demonstrated that the quantitative differences of VOCs in strawberry preserves are related to very slight differences in consumers’ perception and did not induce preferences in the panelists [55].

However, the HS-SPME–GC–MS technique does not give a complete idea of the volatiles released by the sample. For this reason, it is sometimes coupled with other techniques, such as PTR-MS, to gain a more complete idea of the volatile profile of a specific matrix. In fact, PTR-MS has the advantages of not needing the sample preparation step and of a fast sampling time lower than 1 s. In the case of strawberries, Li et al. [52] coupled HS-SPME–GC–MS to PTR-MS to study the changes in the volatile profiles of strawberries harvested at different maturation times during storage.

In agreement with an odor sensing system, an E-nose instrument identifies the differences in the aroma and distinguishes between food varieties on the basis of the aroma patterns. The recommended minimum concentration for the E-nose analysis should not be evaluated on the basis of the type of sensor arrays but on the basis of the odor considered acceptable for the environment [56]. For this reason, the minimum concentration of VOCs is dependent on the type of food, and it ranges from a minimum of 0.01 ppb to a maximum of single-digit ppm [57]. E-noses are good devices to identify food aromas from different sources, as it is able to associate the volatile compound with a food aroma through a digital output [55]. However, an E-nose delivers information on the available compounds in conjunction with a multivariate statistical analysis; an E-nose signifies an effective device for the discrimination of volatile compounds without counting on earlier separation measurements, unlike the existing analytical methods [58,59]. A variety of E-nose sensors are available for agricultural and food product analyses, like for a matrix as strawberries and their features are related to different configuration procedures that involve the response variables and output data [60]. The main components of the E-nose device are a chemical sensor array, a pattern recognition system prepared with a signal processing mechanism, and an odor sample collector that comprehends a volatile response. The sensor detector is generally made from different types of materials, such as metal oxides (MOS), or conducting polymers that will experience a modification in electrical resistance and sensitivity when exposed to volatile compounds [56].

Regarding applying the E-nose to strawberries (Table 3), Xing et al. [61] developed an E-nose system for detecting strawberry freshness at different storage periods, from harvesting day until decay. Their new system consisted of six MOS semiconductor sensors connected to a data acquisition system and a computer with pattern recognition software. The strawberry’s aroma emitted was analyzed during storage, and the E-nose response values were used to develop a cluster analysis and classification models. Subsequently, the GC–MS results proved the feasibility of the selected sensors array for the detection of strawberry freshness [61]. Furthermore, a method combined with E-nose and hyperspectral imaging (HSI) was applied to detect strawberries’ microbial contents and quality attributes during decay by Liu et al. [62]. This study also showed that combining the two nondestructive techniques could help to enhance the evaluation of strawberry quality and safety [62]. Rao et al. [63] used E-nose technology to characterize strawberry fruits with vibrational damage based on their volatile substances (VOCs). They analyzed four groups of strawberries with different durations of vibrations (0, 0.5, 1, and 2 h), and they collected the E-nose signals at 0, 1, 2, and 3 days after vibration treatment. The main results showed the potential use of E-nose technology to detect strawberries that have suffered vibrational damage [60]. Moreover, Radi et al. [64] designed an E-nose composed of 13 MOS for classifying synthetic flavors and developed a PCA for the pattern recognition software. From the results, the aroma patterns of strawberry and jackfruit tended to be mixed [64]. Additionally, Rasekh et al. [62] examined two types of natural and industrial strawberry juices; as a result of this study, E-nose could be used for real monitoring of the volatile components of the food to evaluate the quality and adulteration of the fruit juice with satisfactory results [65]. Recently, Zhang et al. [66] applied the fusion of E-nose and E-tongue sensors based on changes in the odor and taste features to predict the total bacterial count in freshly squeezed strawberry juice during cold storage. The results showed that microbial contamination was related to the changes in the aroma and taste futures of freshly squeezed strawberry juice during cold storage [66]. In conclusion, the application of GC-based methods PTR-MS and E-nose has greatly increased in the strawberry industry due to the promising alternative in the quality and sensory inspection of food products. As a result, future trends in the utilization of advanced E-nose devices will lead to greater abilities of electronic sensing tools, as well as provide inspections of food products using a more consistent procedure [56].

**Table 3 molecules-28-05810-t003:** Recent applications of GC-based PTR-MS and E-nose analytical methods to study the quality and safety of strawberries.

Technique Name	Type of Study	Sensing System	Analytical Method	Pattern Recognition System	Main Goals	Ref.
GC-based methods	- Study of volatile changes in strawberry juices during storage		HS-SPME–GC–MS		- Changes in the volatile profile, mainly related to esters, aldehydes, terpenes, sulfur compounds, and ketones, are influenced by storage conditions and temperature.	[51]
	- Investigation of the volatile profile of three different varieties of strawberries		Headspace solid- phase microextraction–gas chromatography–flame ionization detector (HS-SPME–GC-FID)		- The genotype can influence the volatile profile of strawberries. Geraniol and hexyl hexanoate resulted to be useful to discriminate among different cultivars.	[31]
	- Comparison of the volatile profiles of two cultivars of strawberry		HS-SPME–GC–MS		- Different cultivation methods can affect the volatile profile of strawberries. Soil culture can improve the aroma profile in cultivated strawberry ‘Amaou’ fruits.	[53]
	- Investigation of the influence of freezing and different drying methods on the volatile profile of strawberry and commercial		HS-SPME–GC–MS		- Shade air-dried, frozen, freeze-dried, and oven-dried 45 °C samples retained more of the fruity and sweet aromas of strawberry fruits. - Strawberry jams demonstrated almost a complete loss of esters and alcohols, while terpenes were significantly increased. - Off-flavor and heat-induced volatiles, such as furfural, were generated in jams.	[12]
	- Evaluation of changes in content of VOCs in Sabrosa strawberry, collected at two different ripening stages		HS-SPME–GC–qMS (triple quadropole mass spectrometry)		- The collection times did not influence the fruit quality. - The ripening stage is determinant to meet consumers’ liking.	[22]
	- Identification of VOCs in strawberry preserves to find a possible correlation between the volatile profile and consumer liking		Dynamic headspace–gas chromatography–time-of-flight mass spectrometry analysis (DHS–GC–Q-ToF)		- Nine VOCs have been selected as predictive for consumer liking of strawberry preserves. - 2-methyl-2-vinyltetrahydrofuran gives a fruity, herbal-minty, piney aroma character. - Quantitative differences of VOCs in strawberry preserves are related to differences in consumer’s perception	[55]
	- Identification of strawberry flower volatiles to study their influence in the pollinators behaviors		HS-SPME–GC–MS		- The volatile composition of floral scents of strawberries influenced the behavior of pollinators.	[54]
Proton Transfer Reaction-MS (PTR-MS)	- Investigation of changes during storage in the volatile profile of strawberries harvested at different maturation times		PTR–ToF–MS		- Strawberries harvested at the ¾ red stage had a lower sweetness with respect to the full red ones. - All-red strawberries can reach the highest values of aroma properties after one day of refrigeration.	[52]
Electronic Nose (EN)	- Detection of strawberry freshness at different storage periods	PEN 3, 10 MOS sensors	Combined with GC–MS	PCA, PLS-DA, SVM	- Developed E-nose system holds the advantage of real-time detection for strawberry freshness discrimination.	[61]
	- Detection of microbial content and quality attributes of strawberries during decay	PEN 3, 10 MOS sensors	Combined with near infrared HIS system	PCA	- The combination of the two sensing techniques hyperspectral imaging (HSI) and E-nose can potentially be implemented for the detection of safety and quality of strawberries.	[62]
	- Detection of strawberry fruits with vibrational damage based on their volatile substances	Fox 4000, 6 MOS sensors	Combined with GC–MS	LS-SVM	- The potential use of E-nose technology to detect strawberries that have suffered vibrational damage.	[63]
	- Identification of different fruits’ synthetic flavors, among them strawberry flavor	Designed E-nose, 13 MOS sensors	E-nose only	PCA	- an E-nose for classifying strawberry synthetic flavors was evaluated.	[64]
	- Examination of different types (natural and industrial) strawberry’s fruit juices	Designed E-nose, 9 MOS sensors	E-nose only	PCA, LDA, QDA, SVM	- The potential use of E-nose technology to control the quality of strawberry’s fruit juices was evaluated.	[65]
	- Predict the growth of bacteria in freshly squeezed strawberry juice	Designed E-nose, 10 MOS sensors	Combined with E-tongue	ANOVA, Pearson’s correlation, Growth curve	- It was feasible to predict the growth of bacteria in freshly squeezed strawberry juice using E-nose and E-tongue sensors.	[66]

Overall, GC–MS is the most applied system for the analysis of aroma-active compounds in strawberry fruits. It offers structural information on the detected compounds, allowing to identify the VOC by comparing their retention indexes (RI) and mass spectrum (MS) with reference standards. In addition, GC–MS systems can be used in combination with FID for the semi-quantitative analysis of strawberry aroma-active compounds due to the stability of the latter, giving a more proportional response to the number of organic compounds burnt independently from the chemical structures. In the last decade, the application of EN technology has significantly increased, especially in the food and beverages industries, which is inspired by the sense of smell. Extracted VOCs are transferred to the detecting system, which is made of sensors that cause electronic responses to react with VOCs; those are transformed into digital values. MOS are the most utilized sensor types in the EN analysis of strawberry aroma compounds. However, EN sensing is mainly used for the characterization of the overall aroma pattern of strawberry samples, so it is generally coupled with other analytical techniques, such as GC, which can qualitatively and quantitatively characterize strawberry odor-active compounds. In addition, the PTR-MS technique is being increasingly applied in strawberry aroma studies. This increase in interest is mostly attributable to its various advantages in VOC analysis, including the absence of sample preparation and the possibility to directly quantify compound levels without the need for a calibration standard. Moreover, coupled with ToF mass analyzers, PTR-MS provides a higher sensitivity and higher mass resolution analysis. Beyond these properties, the use of PTR–ToF–MS can provide interesting results in real-time analyses, allowing to monitor the kinetics of aroma-active compounds during processes such as drying [67,68].

## 5. Conclusions

Several concluding remarks highlighting the contribution of this review to the current knowledge can be summarized as follows:-The aroma of strawberries is developed during ripening, and it varies greatly depending on many factors, including the degree of maturity, strawberry cultivar and variety, environmental conditions, storage, and postharvest methods.-The aroma of strawberries consists of a highly complex mixture of compounds, such as furanones, esters, sulfur, and terpenoids compounds.-The impact of drying as a postharvest treatment on the changes of the aroma and sensory properties of strawberries is greatly variable. Oven drying, freeze drying, vacuum freeze drying, microwave drying, shade air drying, and hot air drying were the most common drying methods used for the production of strawberry powders with suitable aromas. Each drying technique has a different effect at varying levels, but in most of the cases, using higher temperatures resulted in greater deterioration of the sensory properties due to the loss of a major part of the esters.-Gas chromatography, electronic nose sensing, and proton-transfer- reaction mass spectrometry were mostly used to analyze and identify the VOCs of strawberry fruits.-Using a combination of drying techniques to minimize the negative effects on the strawberry aroma and flavor could be recommended. More studies must be conducted to better understand the volatile profile of strawberries and to evaluate the impact of hybrid drying on the sensory properties.

## Figures and Tables

**Figure 1 molecules-28-05810-f001:**
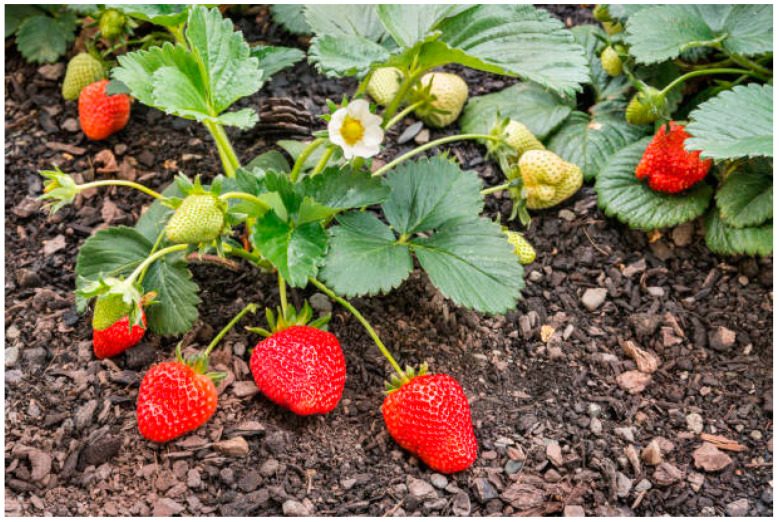
Strawberry plant.

**Figure 2 molecules-28-05810-f002:**
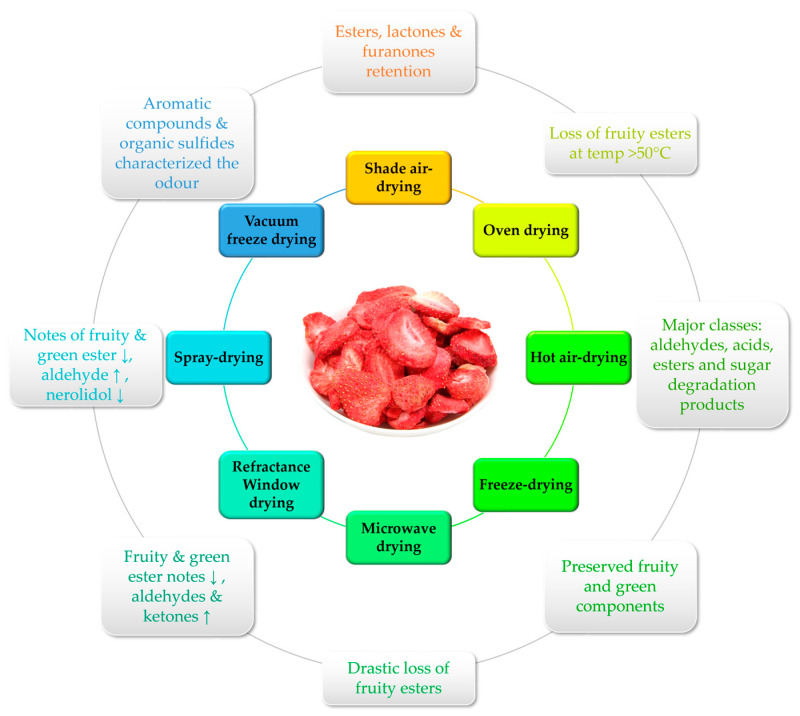
Summary of the impact of different drying techniques on the volatile notes of strawberries.

**Table 1 molecules-28-05810-t001:** The 31 main VOCs identified in red fresh strawberry fruits and their class, odor descriptors, and occurrence in the literature between 2018 and 2023. For each reported reference a specific dot color has been used.

Reference					21	22	23				25	26		6			30										31			12	
	Compound Name	Class	Odor Descriptors	Samples Type	Mean of 6 Genotypes	Candonga Sabrosa	Camarosa	Crystal	Monterey	Portola	Florida Fortuna	Sabrina	Elide	Frontera	Festival	Albion	Rubygem	Fortuna	Festival	Calinda	FL127	Plared	Sahara	Sabrina	Victory	E-22	Camarosa	Candonga	Festival	commercial	Total Occurrence
1	Linalool (3,7-dime-thylocta-1,6-dien-3-ol)	terpens	citrus, fruity/floral		●	●	●	●	●	●	●	●	●	●	●	●	●	●	●	●	●	●	●	●		●	●	●	●	●	25
2	methyl hexanoate (hexa-noic acid, methyl ester)	esters	fruity, pineapple		●	●	●	●	●	●	●	●	●	●	●	●	●		●	●	●		●	●	●	●	●	●	●	●	24
3	nerolidol	terpens	mild floral		●	●	●	●	●	●	●	●	●				●	●	●	●	●	●	●	●	●	●	●	●	●	●	23
4	2-methyl butanoic acid (2-methyl butyric acid)	acids	sour, cheesy, sweety		●	●	●	●	●	●	●	●	●				●	●	●		●	●	●	●	●	●	●	●	●	●	22
5	hexanoic acid	acids	sweety, cheesy		●	●	●	●	●	●	●		●				●	●	●	●	●	●	●	●	●	●	●	●		●	21
6	ethyl hexanoate (hexa-noic acid, ethyl ester)	esters	fruity, sweet, pineapple		●	●	●	●	●	●		●	●				●		●	●	●	●	●	●	●	●		●	●	●	20
7	5-hexyldihydro-2(3H)-furanone (γ-decalactone)	lactones	fruity, peach, sweet		●	●	●	●	●	●	●	●	●				●		●		●		●	●	●	●	●	●		●	19
8	2-hexenal	aldehydes	green, fruity			●					●	●	●				●	●	●	●	●	●	●	●	●	●	●	●	●	●	18
9	ethyl butanoate (buta-noic acid ethyl ester)	esters	fruity, sweet, pineapple		●	●	●	●	●	●		●	●	●	●	●											●	●	●	●	15
10	methyl butanoate (buta-noic acid methyl ester)	esters	sour, cheesy, sweety			●	●	●	●	●	●	●	●	●	●	●											●	●	●	●	15
11	nonanoic acid	acids	waxy, fatty, cheesy			●					●	●	●				●	●	●	●	●	●	●	●	●	●					14
12	octanoic acid	acids	fatty, waxy, rancid vegetable oil			●					●	●	●				●	●	●	●	●	●	●	●	●	●					14
13	benzaldehyde	aldehydes	Sweet, bitter, almond			●	●	●	●	●		●	●							●	●				●	●	●	●		●	14
14	decanoic acid	acids	rancid odor			●						●	●				●	●	●	●	●	●	●	●	●	●					13
15	terpineol	terpens	herbal			●					●		●				●	●	●	●	●	●	●	●	●					●	13
16	acetic acid	acids	strong vinegar								●						●	●	●	●	●	●	●	●	●	●				●	12
17	hexanal	aldehydes	fresh, green			●	●	●	●	●	●	●	●														●	●	●	●	12
18	nonanal	aldehydes	waxy, aldehydic, fatty			●	●	●	●	●	●	●	●	●	●	●														●	12
19	2-hexen-1-ol acetate	esters	pleasant, fruity, green			●	●	●	●	●		●	●				●			●	●				●	●					12
20	benzyl acetate (acetic acid phenylmethyl ester)	esters	sweet, floral, fruity			●							●	●	●	●				●	●	●	●	●	●	●					12
21	hexyl acetate (acetic acid hexyl ester)	esters	fruity, green apple, banana			●	●	●	●	●	●	●	●	●	●	●														●	12
22	DMHF (2,5-dimethyl-4-hydroxy-3(2H)-furanone) o furaneol	furanones	sweet, caramel, candy		●	●	●	●	●	●		●	●														●	●	●	●	12
23	DMMF (2,5-dimethyl-4-methoxy-3(2H)-furanone) o mesifurane	furanones	sweet, caramel		●	●	●	●	●	●		●	●														●	●	●	●	12
24	γ-dodecalactone	lactones	sweet, flower, fruit		●	●						●	●					●	●	●			●	●	●	●				●	12
25	butanoic Acid	acids	sour, cheesy, sweety		●	●					●	●	●				●		●			●	●	●	●						11
26	butyl acetate (acetic acid butyl ester)	esters	fruity, banana			●	●	●	●	●		●	●	●	●	●														●	11
27	hexyl hexanoate	esters	green herbal, fruity			●	●	●	●	●		●	●														●	●	●	●	11
28	farnesene	terpens	woody, green			●	●	●	●	●		●	●	●	●	●														●	11
29	Geraniol (3,7-dime-thylocta-2,6-dien-1-ol)	terpens	sweet, berry, floral														●	●	●		●	●		●		●	●	●	●	●	11
30	1,2-benzenedicarboxylic acid diethyl ester	esters															●	●	●	●	●	●	●	●	●	●					10
31	benzophenone	ketons	balsamic, geranium								●						●	●	●	●	●	●	●	●	●						10

## Data Availability

All reported data are already published, being the present paper a review. Most of references are on Open Access journals.

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
