# Peer review of "Volatile Profile of Strawberry Fruits and Influence of Different Drying Methods on Their Aroma and Flavor: A Review"

_molecules, 2023, doi:10.3390/molecules28155810_

Round 1

Reviewer 1 Report

Authors could find a very detailed Research article on similar lines published already as

Influence of Freezing and Different Drying Methods on Volatile Profiles of Strawberry and Analysis of Volatile Compounds of Strawberry Commercial Jams

published already in same journal in 2021. 

Also other authors have published similar work in the field already.

After two years, there is no new information except for collection of literature.

No novelty or innovation was found.

Authors could find a very detailed Research article on similar lines published already as

Influence of Freezing and Different Drying Methods on Volatile Profiles of Strawberry and Analysis of Volatile Compounds of Strawberry Commercial Jams

published already in same journal in 2021. 

Also other authors have published similar work in the field already.

After two years, there is no new information except for collection of literature.

No novelty or innovation was found.

Reviewer 2 Report

In this review, titled “Volatile Profile of Strawberry Fruits and Influence of Different Drying Methods on their Aroma and Flavor” the authors aims to consider a good reference for research concerning the aroma profile of strawberries.  It helps to better understand the complex aroma and flavor of strawberry and provide a guide for the effects of drying processing. Overall, this is an interesting paper that provides promising data for an important issue of the food science and modern industries research. In my opinion, the study is need minor revision to be suitable to publish due to the following issues,

1- The manuscript is well written, but it needs some revision because there are some mistakes and some sentences are not clear.

2- Should add image for Strawberry plant and Botanical Voucher Specimen, ↑MOBOT, Tropicos.org in introduction section

3- In the introduction section should more discus about Strawberry properties not just one paragraph

4- One of the most important things in this review is the effect of the offshoots growing, climate change, environment, soil, and the water used in strawberry cultivation, and its effect onto the chemical compounds content, which have a major role in distinguishing the taste and aroma of the final product of the crop. Should discuss and add it

5-  The resolution of Figure 1 is so low so please modification it.

Minor editing of English language required

Author Response

I attached the coverletter.

Reviewer 3 Report

This work reviews relevant literature on volatile organic compounds from strawberry fruits, as well as on the influence of different drying methods on their aroma and flavour.

 Unfortunately the manuscript has several draw-backs, hence please consider the following suggestions for an improved version:

- English needs improvement

L.48 – define the acronym HS–SPME–GC–MS before its first use;

L.49-50 – delete “In the current studies, HS–SPME–GC–MS was used to determine and analyse the volatile profiles of strawberries” duplicates the previous text

L.50-51 – delete “At the same time, the ability of the E-nose to distinguish different volatile profiles was evaluated.” – already mentioned in L.47

L.70 – delete subtitle – this is the only one in this section, hence is useless; eventually, you can start the introductory section with these two paragraphs, but you have to shorten them, since the content is off-topic

L.103 – delete “volatile organic compounds” – acronym defined in L.39

L.106 – 107 – delete “This chapter explores the volatile compounds responsible for the fresh strawberry aroma.” – obvious, see the title/ filler

L.177 – rephrase “defined” – not appropriate in this context

L.210 – delete “volatile organic compounds” – acronym defined in L.39

L.259 – rephrase “addiction” – inaccurate technical language

L.365 – from figure 1, a reader can conclude that following a cyclic pathway, the mentioned effect can occur, which is not true; besides, there is no “entrance point” in this cycle. Find another graphical approach or better use a simple table

L.367 - Table 2 – too wordy, resume the information in smaller text blocks (seems to be the result of copy/paste from the cited paper) / rephrase “addiction” – inaccurate technical language

L.368 – when reviewing analytical methods, it is expected to give an appropriate amount of critical opinion on these, emphasising at least advantages/ limitations then comparing the reviewed methods – consider adding these issues

L.369 – delete “volatile organic compounds” – acronym defined in L.39

L.415 – 416 – what is the meaning of “real-time idea…” in this context? Consider rephrasing

L.422 - delete “electronic nose” – acronym defined in L.422

L.444 - delete “metal oxide” – acronym defined in L.440

L.448 - delete “gas chromatography-mass spectrometry” – acronym defined in L.46

L.460 - delete “principal component analysis” – acronym defined in L.233

L.475, Table 3 – delete all instances of  “Headspace-solid phase microextraction-gas chromatography-mass spectrometry” – acronym defined in L.375. This table is wordy too, as Table 2, hence delete all unnecessary words,  such as “results showed that ”, “in particular”, “in addition”, “this study illustrated that ”, etc./  What is the meaning of PEN 3, 10…

L.477 – replace the current text with several concluding remarks highlighting the contribution this review has to the current knowledge, avoiding the re-iteration of known issues

English needs moderate improvement

Round 2

Reviewer 1 Report

I dont find anything new and contribution to existing information.

As conveyed earlier that there is plethora of data on said topic apart from what authors published already in form of research article.

Author Response

R: We confirm that the idea of the review is totally new as stated before. No previous reviews provided the same data mentioned in the present paper, such as summary of the relevant literature focusing on the active aroma components in strawberry and the many analytical methods used to identify them, including gas chromatography, electronic nose sensing, and proton-transfer reaction mass spectrometry. More, no previous review discussed the Influence of different drying methods on Volatile Profiles of strawberry, as this review does.

Reviewer 3 Report

Despite the level of the manuscript increased after the first review, there are still several issues to address:

-       tables 2 and 3 are unnecessary wordy; keep only key facts here and  better use bullet lists than classic phrases;

-       my former recommendation related with the “Conclusions” was “to replace the current text with several concluding remarks highlighting the contribution this review has to the current knowledge, avoiding the re-iteration of known issues”; instead, the authors added more text! Keep in mind that this is a concluding section, hence discussions are not appropriate here; try to summarize this section in 3-4 concluding remarks.

Only minor editing of English language are required

Author Response

Despite the level of the manuscript increased after the first review, there are still several issues to address:

-       tables 2 and 3 are unnecessary wordy; keep only key facts here and  better use bullet lists than classic

phrases;

R: Thank you to the reviewer for the comment. We reduced the number of words again in Table 2 and 3, and we used  bullet lists according to reviewer suggestion.

-       my former recommendation related with the “Conclusions” was “to replace the current text with several concluding remarks highlighting the contribution this review has to the current knowledge, avoiding the re-iteration of known issues”; instead, the authors added more text! Keep in mind that this is a concluding section, hence discussions are not appropriate here; try to summarize this section in 3-4 concluding remarks.

R: Thank you to the reviewer for the comment. We summarized the Conclusions section in 5 concluding remarks according to reviewer suggestion.

Comments on the Quality of English Language

Only minor editing of English language are required

R: English was improved according to reviewer suggestion.